# Convergence and equilibrium in molecular dynamics simulations

Franco Ormeño[1] & Ignacio J. General 🔵 [2✉]

Molecular dynamics is a powerful tool that has been long used for the simulation of biomolecules. It complements experiments, by providing detailed information about individual atomic motions. But there is an essential and often overlooked assumption that, left unchecked, could invalidate any results from it: is the simulated trajectory long enough, so that the system has reached thermodynamic equilibrium, and the measured properties are converged? Previous studies showed mixed results in relation to this assumption. This has profound implications, as the resulting simulated trajectories may not be reliable in predicting equilibrium properties. Yet, this is precisely what most molecular dynamics studies do. So the question arises: are these studies even valid?Here, we present a thorough analysis of up to a hundred microseconds long trajectories, of several system with varying size, to probe the convergence of different structural, dynamical and cumulative properties, and elaborate on the relevance of the concept of equilibrium, and its physical and biological meaning. The results show that properties with the most biological interest tend to converge in multi-microsecond trajectories, although other properties–like transition rates to low probability conformations–may require more time.

[1] Instituto de Investigaciones Biotecnológicas, Universidad Nacional de San Martín, San Martín, Buenos Aires, Argentina. [2] Escuela de Ciencia y Tecnología, Universidad Nacional de San Martín, ICIFI and CONICET, San Martín, Buenos Aires, Argentina. ✉email: igeneral@unsam.edu.ar

The analysis of Molecular Dynamics (MD) simulations is most often based on the implicit assumption that the system was in thermodynamic equilibrium. For example, in biomolecular simulations, the starting point is usually an experimentally determined 3D structure, retrieved from the Protein DataBank[1] that is not in equilibrium, since its determination required specific preparations (for x-ray diffraction, the most usual experimental technique used for structure determination, the biomolecule is part of a crystal from where electron density maps are obtained; these are strong non-equilibrium conditions, if one wants to simulate a physiological–non-crystal–system). Consequently, the typical simulation protocol consists of the energy minimization of the system, followed by the so-called equilibration steps, where the system is first heated and pressurized to the target values, and then a relatively long and unrestrained simulation is carried on, to let the system explore its conformational plus velocity space (i.e., the phase space) and relax, reaching thermodynamic equilibrium. But here lies an often oversaw detail: how can we determine if the system reached true equilibrium?

A standard way to check for equilibration is to plot several magnitudes calculated from the simulation, as a function of time, and see if they have reached a relatively constant value (a plateau in the graph). Simple and often used metrics are the energy and the root-mean-square deviation (RMSD) of the biomolecule. The latter is supposed to reach a plateau when the potential energy of the biomolecule reaches a minimum (although this is not necessarily true). There are more sophisticated ways of checking for equilibration, but they tend to be more complex and time-consuming, so they are usually not employed[2]. One of those other ways was used by Hu et al.[3], who studied time-averaged mean-square displacements and autocorrelation functions (ACF) of some properties of three biomolecules, extracted from long MD simulations, noticing that convergence was not achieved, and concluding that, in general, biomolecules do not reach equilibrium, even in simulations on the order of tens of μs. Furthermore, they proceeded to study ACFs of two more biomolecules, from results of single-molecule spectroscopy experiments, and concluded that some proteins may posses non-equilibrium behavior for times longer than hundreds of seconds. This is a very strong conclusion, that directly affects the relevance of MD studies. Is MD a valid tool to analyze the equilibrium properties of proteins? Or is it highly insufficient, since the time-ranges that are currently possible to simulate are very, very far from the needed ones? This is a question that, in our opinion, is being surprisingly ignored by the community, but needs to be thoroughly addressed since, if Hu et al.'s conclusions are generally true, then a majority of currently published MD studies would be rendered mostly meaningless. Strangely enough, as the question is ignored and publications using MD are very common and are presented as if their results are clearly converged, at the same time there is a sort of acceptance of Hu et al.'s conclusion, with many authors citing their result as common knowledge[4–8] but none (to the best of our knowledge) testing, extending or replicating those results.

This non-equilibrium behavior was also described in relation to surface water around proteins; using neutron-scattering experiments and MD simulations, Tan et al.[9] found sub-diffusive motion of water molecules during time-intervals of around 1 ns; but for longer windows, of about 100 ns, the sub-diffusivity tended to disappear. Interestingly, using a Go model of a λ-repressor protein, Krivov[10] found that subdiffusivity appeared when a non-optimal reaction coordinate was employed, but it disappeared when switching to an optimal one, pointing to the importance of a careful choice of the property to be measured.

From a Statistical Mechanics point of view, the physical properties of a typical system are derived from its conformational

partition function, Z, i.e., the volume of the available conformational space, Ω, weighted by an exponential factor of the energy (see Eq. (1) for the expression of Z in the Canonical ensemble; the exponential, or Boltzmann factor, represents the probability of observing a state with energy $E(\mathbf{r})$ in Ω). Z should have the correct contribution from all physically allowed conformations, including low probability ones. On the other hand, when calculating the average of a property A, $\langle A \rangle$ (such as a distance, angle, RMSD, etc), the mathematical expression can be written as an average of the values of the property, weighted by the Boltzmann factor (Eq. (2)). But low probability regions of the conformational space will not contribute much to $\langle A \rangle$, since they tend not to happen and, hence, $\langle A \rangle$ does not require a full exploration of Ω; counting contributions only from the most probable regions can lead to a very good approximation.

$$Z = \int_{\Omega} \exp\left(-\frac{E(\mathbf{r})}{K_B T}\right) d\mathbf{r} \tag{1}$$

$$\langle A \rangle = \frac{1}{Z} \int_{\Omega} A(\mathbf{r}) \exp\left(-\frac{E(\mathbf{r})}{K_B T}\right) d\mathbf{r} \tag{2}$$

$$F = -K_B T \ln(Z) \tag{3}$$

$$S = -\left(\frac{\partial F}{\partial T}\right)_V \tag{4}$$

On the contrary, properties like transition rates to and from unlikely regions of Ω depend explicitly on the probability values of those regions and, thus, do require their thorough exploration. In this sense, a system can be in *partial equilibrium* so that some properties have already reached their converged values, while others have not. This is not the standard physical definition of the concept of equilibrium, where one studies a *perfect* thermodynamic equilibrium (full exploration of Ω) but, as argued later, it may be appropriate for biomolecular systems and, specifically, for MD applications. This point is suggesting that free energy and entropy–the fundamental physical magnitudes directly related to equilibrium–may not be ideal metrics in the present case. The reason is that the statistical mechanical expressions for those quantities (Eqs. (3–4)) depend explicitly on the partition function, and are not formed as averages, so they need to keep all contributions from the conformational space, including low probability ones. Consequently these quantities cannot, in principle, be separated into partial contributions of a given region or a specific motion of the protein and, thus, cannot be used to study partial equilibrium.

Guided by the these ideas, we give here a clear working definition of the concept of equilibrium, as applied in this study to MD simulations: "Given a system's trajectory, with total time-length T, and a property $A_i$ extracted from it, and calling $\langle A_i \rangle(t)$ the average of $A_i$ calculated between times 0 and t, we will consider that property "equilibrated" if the fluctuations of the function $\langle A_i \rangle(t)$, with respect to $\langle A_i \rangle(T)$, remain small for a significant portion of the trajectory after some "convergence time", $t_c$, such that $0 < t_c < T$. If each individual property, $A_1, A_2, ...$, of the system is equilibrated, then we will consider the system to be fully equilibrated".

This definition makes a clear distinction between the concepts of *partial* and *full* equilibrium, in preparation to find systems where only some properties have reached convergence, but others have not. This is precisely what is needed when calculating, for example, a distance between two domains of a protein (an average property that depends mostly on high probability regions of Ω), and the free energy of the entire protein (a property that depends on all regions of Ω). But there are two main problems with this definition: (1) The meaning of the phrase *"remain small for a*

*significant portion of the trajectory"* is vague and relative, as the fluctuations should go to zero as the thermodynamic limit is approached, but MD simulations are far from reaching it. MD ensemble sizes are small, due to the short times usually simulated (very long runs for today standards–on the order of milliseconds–still represent ensembles very far from the thermodynamic limit). (2) $\langle A_i \rangle(T)$ should be changed to $\langle A_i \rangle(t = \infty)$, the infinite-time average. But these averages can only be known experimentally (or theoretically in an analytical case); hence, in the event where it is unknown, there is no way to clearly establish how far a finite-time average is from the infinite-time one. And even if a known value is available, simulated systems are seldom true representations of experimental ones, as the conditions in which they are carried out are different and, thus, their averages may be different (although good models and simulations should keep the difference to a minimum). So, in order to make sense of the definition, we will take $\langle A_i \rangle(T)$ as the "correct value". These 2 problems make the estimation of the convergence of properties more of an art than a precise determination. Even in an ideal case, where some property is found to approach a fixed value, with small fluctuations, it is not possible to affirm that it will stay at that value if the simulation is continued; the system may be stuck at some deep local minimum of energy, from which it could eventually escape in a longer simulation.

Keeping the previous caveats in mind, here we examine the convergence of multi-microsecond simulations of several proteins, by analyzing different metrics. We also include autocorrelation functions, and try to discern what they tell about the equilibrium of a system, by first studying them in analytical cases.

## Results

**Dialanine - Unconverged properties in a mostly converged system.** The first system analyzed was dialanine, a very simple, 22-atom toy model of a protein. It is reasonable to hypothesize that, due to its very small size, this molecule would reach equilibrium within the usual time-lengths used in MD simulations, making it an ideal study-case[11,12]. On the other hand, if equilibrium is not reached, it could then be assumed that it will neither be generally attainable for larger, more complex proteins. With this in mind, Fig. 1 presents structural results from the analysis of a 20 μs trajectory. It shows the distributions of the $\psi$ and $\phi$ dihedral angles as a function of time, aligned with a plot of the free energy landscape in terms of those two angles. $\psi$ populates two main regions, around $(-50, 50)$ and $(120, 190)$, with a very high frequency of transitions between them, while $\phi$ also visits two regions, $(-180, -50)$ and $(30, 80)$, with the latter being barely explored. This is clearly observed in the free energy plot, where six regions are visible (the most populated $\phi$ region appears broken into two sub-regions), but the two corresponding to the positive values of $\phi$ have a greater free energy and, thus, lower probability. Notice the very fast transitions between $\phi$ regions, where no intermediate micro-configurations were observed, as opposed to the $\psi$ ones, with many events clearly visible in the graphs.

This is indicating that $\psi$ has two comparable minima in the $\psi$ subspace (the $(120, 190)$ is the global minimum, by about 1 kcal/mol), but the barrier between them is not high enough ($\sim$3 kcal/mol) to disallow a very large number of transitions between the two. On the other hand, the $\phi$ regions are much more isolated from each other, with significantly different minima (by about 3–4 kcal/mol) and larger barriers ($\sim$5 kcal/mol).

In terms of transition rates, there is a clear difference between the two angles: $\psi$ stabilizes very quickly, reaching convergence in a small fraction of the whole trajectory, but $\phi$ takes a much longer time to achieve convergence, and it could be concluded from the plots that it does not reach it. This is an illustration of the

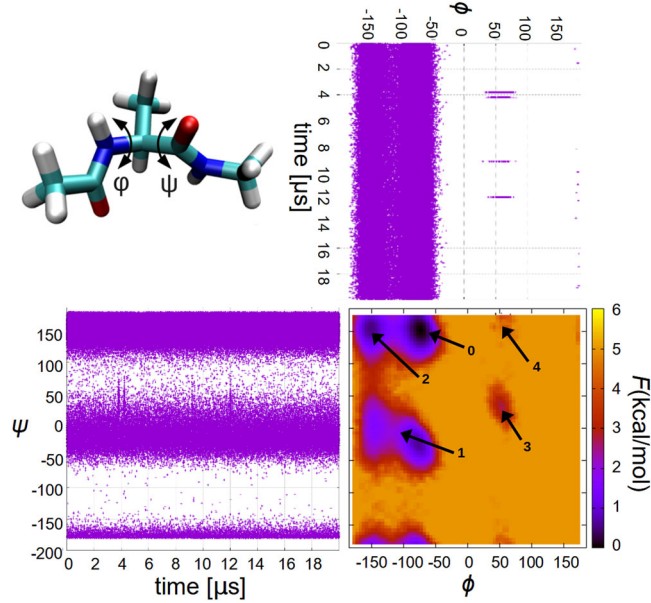

**Fig. 1 Free energy and dihedral angle transitions in dialanine.** Structure of dialanine with definition of dihedral angles $\psi$, $\phi$, along with their time evolution and free energy landscape (with labeled minima).

**Table 1 Metrics used to study equilibrium.**

| | |
|---|---|
| GLOBAL DYNAMICAL CONVERGENCE | Partial components (PCs) |
| | PCs cumulative overlap (CO) |
| LOCAL DYNAMICAL CONVERGENCE | Atomic fluctuations (RMSF) |
| STRUCTURAL CONVERGENCE | Cluster analysis |
| CUMULATIVE CONVERGENCE | Autocorrelation functions (ACF) |

well-known fact that different mechanisms in a given system can have, and usually do have, very different convergence times.

Is this indicating that the trajectory did not achieve convergence and, thus, no equilibrium properties can be obtained from it? In fact, equilibrium properties of dialanine can be clearly extracted, such as the aforementioned transition rates between $\psi$ regions (see Table 4), or any other property that does not depend on the exploration of the $\phi$ sub-space, such as the relative energy or free-energy of the four most stable macro-states (those with $-180 < \phi < -50$). As previously mentioned, in order to accurately calculate equilibrium properties, a simulation should be allowed to thoroughly explore all of the system's available phase space, including all local minima in the rugged landscape of a complex system. But if a local minimum is very unlikely–like the $\phi \sim (30, 80)$ region–then its contribution to the partition function and its derivable properties, will be very small and possibly negligible, making the calculation a very appropriate one.

The results related to the other metrics mentioned in Table 1 are analyzed in the Supplementary Information, and displayed in Supplementary Fig. 1. They all show very good convergence, starting with time-lengths as short as tens of ns. A movie of the trajectory of dialanine and of all the other proteins studied can be found in Supplementary Movies 1–8, along with pdb files containing first and last frames of each simulation, as Supplementary Data files 1–16.

**PGK.** Phosphoglycerate kinase (PGK) is the 415 residues kinase studied by Hu et al., that inspired the present work. Its

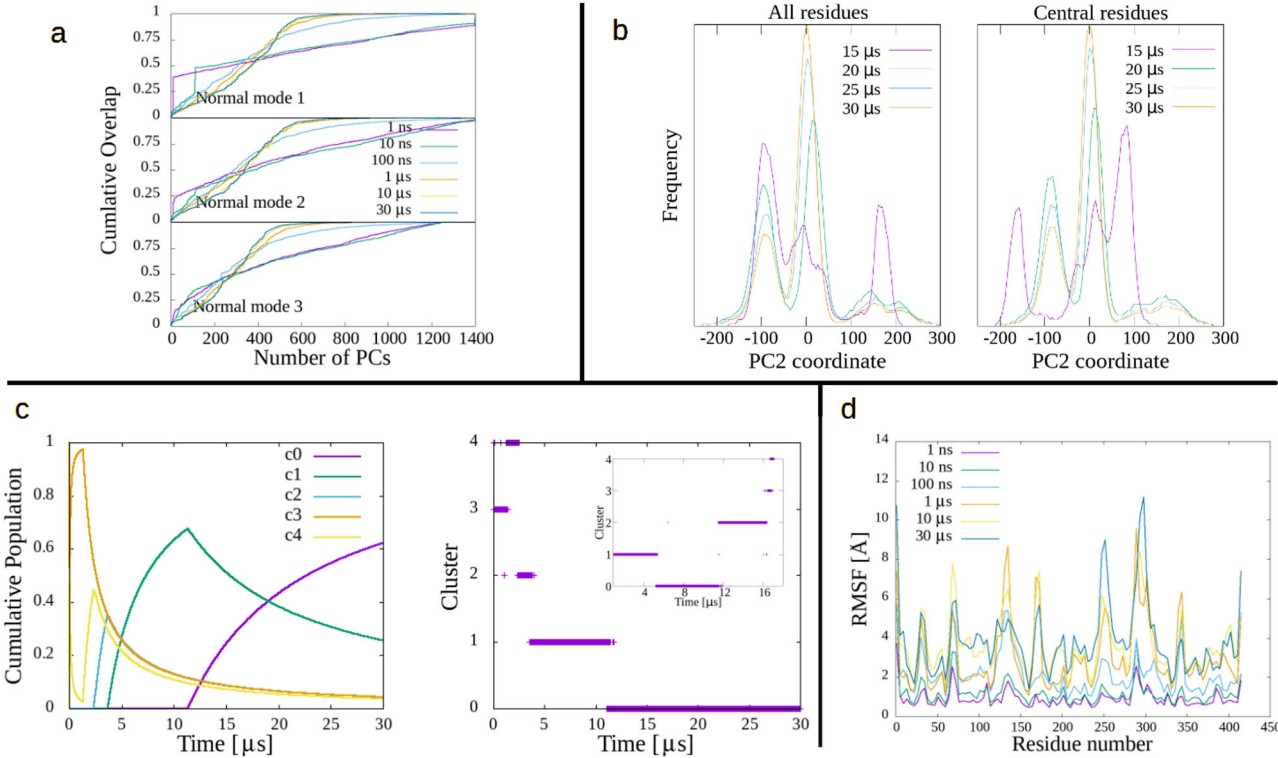

**Fig. 2 Metrics of convergence in PGK. a** Cumulative overlap of first 3 three ANM modes, in terms of MD PCs, for several simulation times. **b** Frequency distribution of selected PC for several simulation times. **c** Cluster cumulative population (left) and cluster time-distribution (right). The main graph on the right shows clusters of the entire 30 μs simulation, while the inset considered only the last 17 μs of it, showing a sub-structure of cluster 0 from the main graph. **d** Residue root-mean square fluctuations for several simulation times.

corresponding results are presented in Fig. 2. CO and RMSF (panels A and D) show good convergence as the length of the MD increases, and panel C also represents good convergence. Meanwhile, panel B appears to somewhat question this, since PC2 (although the same conclusion is valid for the rest of the first 5 PCs) reveals that the population of its two highest maxima (around $-100$ and $0$ Å) is still changing in the last two time lengths. While the cluster distribution in panel C shows C0 being the only cluster in the last half of the trajectory, a cluster study of only this part (last 17 μs)(inset) shows several clusters; this can be interpreted as a subcluster distribution of C0 (of the 30 μs cluster distribution). This explains that once the system settles in cluster 0, at around 11 μs, it is still going through other internal transformations which bring up this substructure, generating variations in the PC2 distribution (the same happens with the other global PCs, not shown).

**SARS-CoV-2 3CL$^{pro}$**. The 3C-like proteinase (3CL$^{pro}$), also known as the nonstructural protein 5 (nsp5), is the main protease of coronavirus. SARS-CoV-2 3CL$^{pro}$ is a dimer of 302 aminoacids per chain which, in its apo state (pdb code: 6Y84), was simulated for 100 μs[13], making it the longest trajectory in the present study. The results, displayed in Fig. 3, show a good convergence of the simulation, just like in previous cases, particularly panels A, C and D. Panel B illustrates an interesting point, as PC1 shows a clear difference in its distribution when going from the 10 to the 50 μs simulation (general shape somewhat similar, but with a different location of the peak), whereas PC2 shows a much quicker convergence, with those two times having practically the same distribution, even of the peak. In both cases, the 1 μs case appears far from convergence.

Panel C reveals a distribution of clusters in relative equilibrium but, at the end, around 90 μs, a new cluster emerges. It appears to correspond to a localized conformational change, given by a 7 aminoacid-long region, which folds into a helix at this time. Visually (see Supplementary Movie 3), this is a minor change that does not appear to perturb the rest of the molecule. But, unfortunately, there is no way of knowing if it would not affect the system in a longer run, past the 100 μs. This is a point that should always be kept in mind when analyzing MD trajectories.

**Other systems**. The other systems appearing in Table 3 and not mentioned so far, Trp-Cage, VHP, GAAC, Barnase and Elastase, are analyzed in the Supplementary Discussion (see Supplementary Figs. 2, 3, 4, 5, 6). But summarizing their results, it was found that they behave very similarly to the ones already presented, showing a good convergence in all metrics, as the simulation times increases, with mostly very good convergence of properties after times on the order of the microsecond. Also, all initial and final frames of the MD simulations are available as Supplementary Data 1–16.

**Autocorrelation functions**. In order to directly compare the results of the present study with those presented by Hu et al.[3], we calculated the ACFs for different properties of each of the studied systems. Figure 4 shows the Decorrelation Curves (DC), i.e., characteristic decorrelation time ($\tau_c$) of the eight studied proteins, as a function of the simulation time (t), in a log-log scale. It is clearly seen how, for all systems, $\tau_c$ increases with the simulation time–an expected fact, as the simulations tend to reach an equilibrium state–until log(t) ~ 7, where a convergence in $\tau_c$ seems to be appearing in every case, except for dialanine and GAAC, where

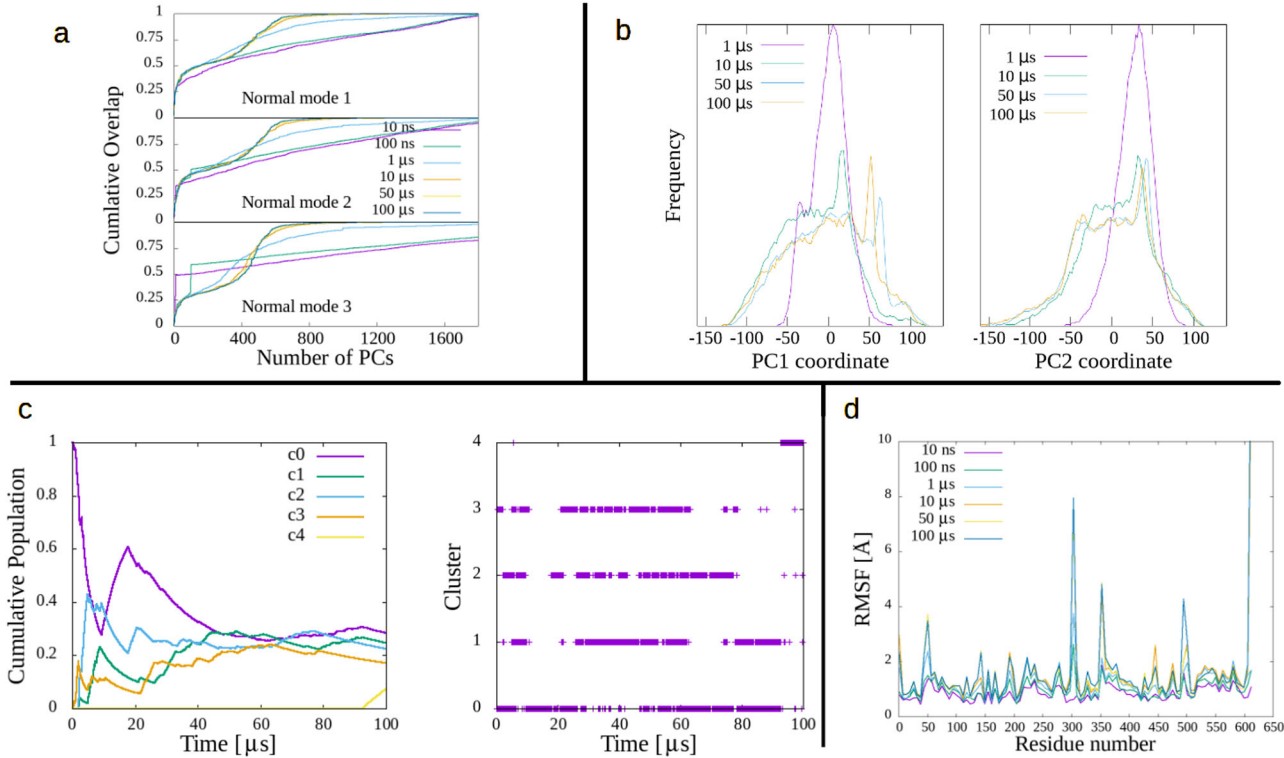

**Fig. 3 Metrics of convergence in 3CL$^{pro}$. a** Cumulative overlap of first 3 three ANM modes, in terms of MD PCs, for several simulation times. **b** Frequency distribution of selected PCs for several simulation times. **c** Cluster cumulative population (left) and time-distribution (right). **d** Residue root-mean square fluctuations for several simulation times.

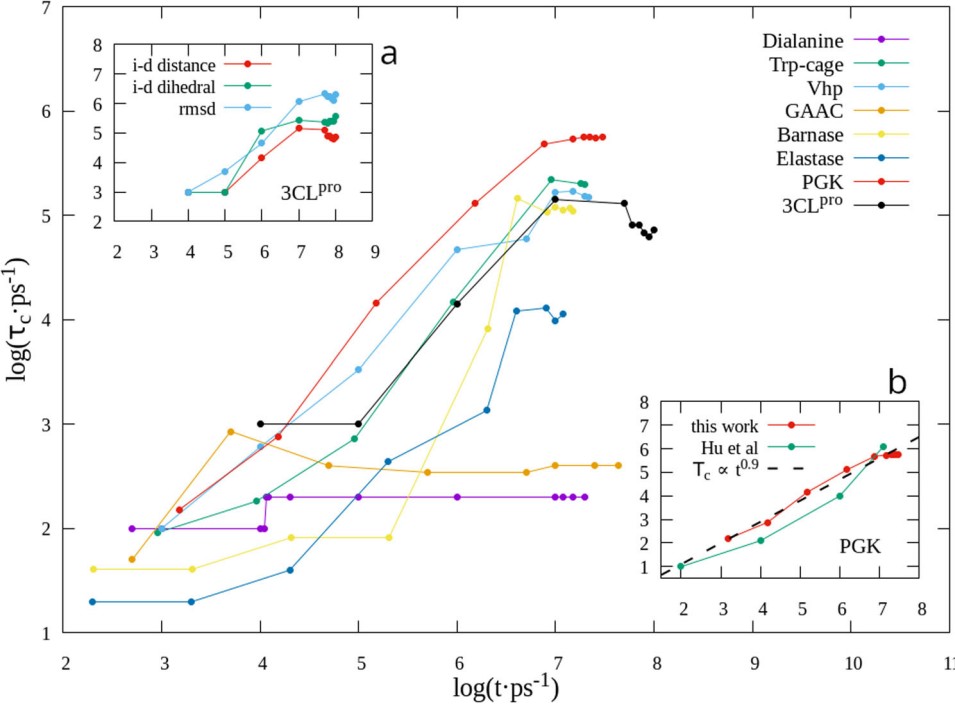

**Fig. 4 Decorrelation curves of protein systems.** The studied magnitudes are: dihedral angle $\psi$ (dialanine), end-to-end distance (Trp-cage, Vhp, GAAC), inter-domain separation (Barnase, Elastase, PGK, 3CL$^{pro}$). **a**: Decorrelation curves for three properties of 3CL$^{pro}$. **b**: detail of PGK's decorrelation curves, from this work and from Hu et al.[3].

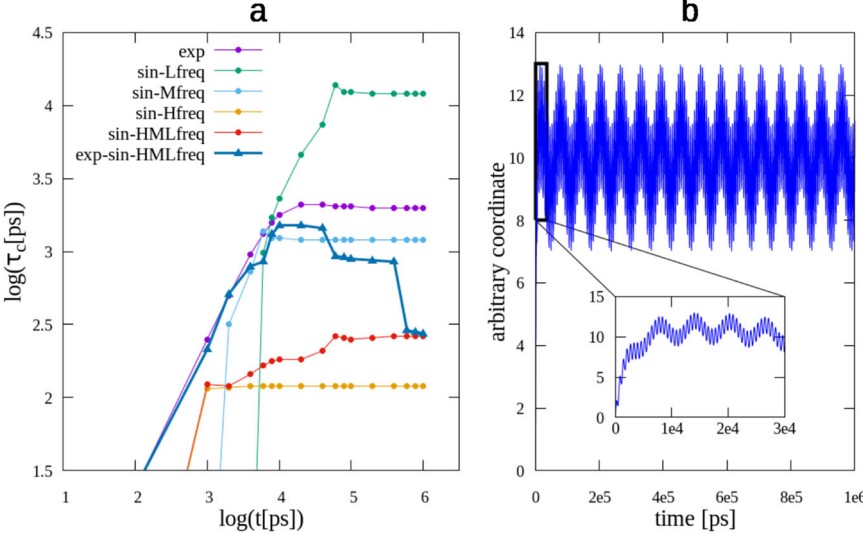

**Fig. 5 ACF of a composite function. a** shows the decorrelation vs the observation time, in a log-log scale. **b** shows the composite function being analyzed, $f(x) = A − A \cdot exp(−B \cdot t) + sin(C \cdot t) + sin(D \cdot t) + sin(E \cdot t)$ (see text for specific value of coefficients), with a zoomed-in region that displays the high-frequency sine component.

| Table 2 Analytical systems used to check convergence: $f(x) = A − A \cdot exp(−B \cdot t) + sin(C \cdot t) + sin(D \cdot t) + sin(E \cdot t)$. | | | | | |
|---|---|---|---|---|---|
| **Function type** | **A** | **B** | **C** | **D** | **E** |
| Exp | 10 | 0.0005 | 0 | 0 | 0 |
| Sine-L | 10 | 0 | 0.0001 | 0 | 0 |
| Sine-M | 10 | 0 | 0.001 | 0 | 0 |
| Sine-H | 10 | 0 | 0.01 | 0 | 0 |
| Sine-LMH | 10 | 0 | 0.01 | 0.001 | 0.0001 |
| Exp-Sine-LMH | 10 | 0.0005 | 0.01 | 0.001 | 0.0001 |

convergence occurs before. Inset panel a in the figure depicts the DC for three different properties of 3CL$^{pro}$: interdomain distance (separation between center-of-mass, CoM, of each domain), interdomain dihedral (taking the CoM of four helices, two in one domain, two in the other), and RMSD of the whole protein. They are qualitatively similar, but as different between them as the graphs of other proteins. This shows, once again, how different types of motion (or different modes) of the same protein can behave quite differently.

To better understand some details of ACFs, Fig. 5 displays the DCs of a few analytical systems, all described by the same function–an exponential with three sines added–each with varying parameters, so that the different parts of the function are included or not, and made to converge with different speeds. The coefficients are specified in Table 2. Panel b in the Figure displays the full function, with all non-zero parameters. In the main graph, the low frequency sine is clearly seen, while the mid frequency one is partially visible; the exponential and fast frequency sine are not observable at all. On the other hand, the inset in this panel shows a fast initial increase, due to the exponential part, and the oscillations of the high and mid frequency sine functions; here, it is the low frequency sine that is not visible. Panel a of the Figure shows the DCs of all the combinations of parameters in Table 2, with the thick line representing the full function. It is interesting to see how a composite function is guided by the DCs of its components. This is clear in the case of the sine with the three frequencies (sin-HMLfreq); it shows a first convergence attempt at $log(t) \sim 3$, where sin-Hfreq converges, followed by a second attempt around 4, where sin-Mfreq converges, and it finally fully converges

around 5, where sin-Lfreq also plateaus. The full function, including the exponential part, also shows this *guidance* by its components, although the exponential garbles the interpretation, since its convergence takes longer than the mid and high frequency sines, partially hiding their effects. But still, it shows 4 convergence attempts, with the last one coinciding with the previous no-exponential function (since the exponential is completely converged at that final time). In summary, this analytic system shows that the ultimate convergence of a composite function's DC is subjected to the convergence of its slowest converging part.

An interesting conclusion can be drawn from this analysis, regarding the time it takes for an oscillation to reach convergence. The motion with the longest convergence time is the low frequency sine, sin-Lfreq, with a period of 63,000 ps. Its DC's convergence time is close to 100,000 ps, which is about 1.5 times its period. Thus, a property characterized by an oscillation with period T could be expected to necessitate a simulation of about $1.5 \cdot T$, in order to reach convergence. This may sound short, as intuitively one might have expected to need several full oscillations before reaching equilibrium, but this is not the case for an ideal sine motion, and may be used as a guide for other cases.

## Discussion

**Functional dynamics converges in the tens of $\mu s$ time-scale of MD.** The central question this work aims to answer, is if proteins simulated via MD–in today's usual time ranges of hundreds of ns to a few tens of $\mu s$–reach equilibrium. But this is a tricky question, since proteins are complex objects, composed of thousands of atoms forming domains and sub-domains, with many different processes contributing to their relaxation (interactions with neighboring molecules, rotation of side-chains, linear and angular oscillation of bonds, large conformational changes, etc), each with its own characteristic time. These motions can be modeled by the harmonic normal modes, which are, in a sense, the most fundamental types of motion of the system. But therein lies the crux of the problem: each mode has a defined and different natural frequency. Hence, at any given time, some modes may be equilibrated while others are not. Although, physically speaking, one should avoid talking about equilibrium in such a case, biologically

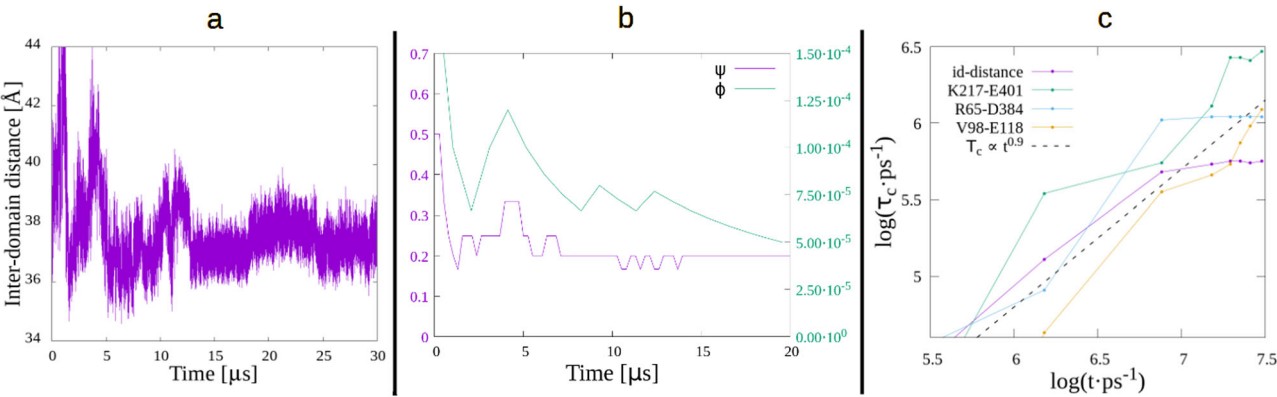

**Fig. 6 Some conclusions. a**: PGK's inter-domain distance. **b**: Dialanine transition rates for dihedral angles $\psi$ and $\phi$ (left and rigth axes, respectively). **c**: DC for several residue-residue distances in PGK.

it makes sense to do so, since those converged modes may be the ones relevant for a given property of the protein. For example, as mentioned earlier, there is this emerging paradigm in molecular biology of structure governing dynamics governing function, and it has been well studied[14–18] that the dynamics involved with function is mostly the one described by the global normal modes. Therefore, it stands to logic that a trajectory where these modes have converged, represents a system that has reached equilibrium in terms of functional dynamics. Yet, in this work several other metrics were tested, including global dynamical convergence, but also local, structural and cumulative convergence (Table 1). All of them showed very good convergence at the longest simulation times, on the order of the few $\mu$s, with the DCs converging a little later, around 10 $\mu$s for most systems. Since the studied proteins had different sizes, up to 612 residues, we may conclude that other systems in this range are also likely to find convergence in their properties, in $\mu$s-long MD simulations.

Is this conclusion disagreeing with that of Hu et al.[3], that *proteins are non-equilibrium and self-similar* for much longer times than these? In terms of numerical results, comparing the DCs for PGK (inset panel b in Fig. 4), they appear to be very similar: the dashed line is the power-law that Hu et al., showed to fit the behavior of the studied proteins. But when looking with more detail at the top region of the curve ($\log(t \cdot ps^{-1}) \sim 7$), it is seen that it starts to show convergence; plotting only one point, as done in the cited work, hides this convergence. In conclusion, in the region of the DC for which there is data, Hu's and this work's curves coincide very well (as they should, since they are both based on the same data), but the difference appears in the extrapolation that the curve will continue following the power law; here we found that convergence starts to appear at the end of it, so the power law prediction breaks. It should be mentioned that this power law describes sub-diffusive dynamics; this is represented by the $\tau_c \sim t^{0.9}$ dashed line of the Figure. Several studies[3,19–21] have analyzed this phenomenon in protein dynamics, finding many examples of it, and concluding that the mechanisms that give rise to sub-diffusivity are varied (fractal topology of the energy landscape, trapping models, fractional Brownian motion, etc). The phenomenon is also present in our study, but it ends in all studied cases–at least in relation to the inspected properties–and is replaced by a stationary or converged value of that characteristic.

But our results also appear to contradict other studies. Yang et al.[22] did a single-molecule electron-transfer (ET) experiment to measure the ACF of the fluorescence-lifetime-fluctuation of a flavin reductase (Fre) in complex with a flavin adenine dinucleotide (FAD). They argued that the measurement is directly related to the ACF of the distance between FAD and

Fre's Tyr35, resulting in a point in its DC with coordinates (14.5, 10.5). Also, Min et al.[23] measured the distance fluctuations between a tyrosine in a monoclonal antifluorescein (Anti-FL) and a fluorescein (FL), via photoinduced ET. The resulting point in a DC graph is (14.5, 12.0). These points from two different experimental works, if included in Fig. 4, would fall close to Hu's fit, $\tau_c \sim t^{0.9}$, apparently implying that the two systems are not converged for times up to the order of a hundred seconds. But it must be stressed that both studies measure the ACF related to the distance between a small molecule (FAD or FL) and a specific tyrosine; this is a local measurement, that takes into account only a few atoms, and is qualitatively different from the cases considered in our study like, e.g., the inter-domain distance in PGK. As previously argued, the latter is a *global* property, the kind that is thought to be involved in the functional dynamics of the protein. As an example of local measurements in one of our proteins, PGK, panel C in Fig. 6 shows the DC for a few residue-residue distances: K217-E401 and V98-E118 show no convergence and loosely follow the power-law, while R65-D384 behaves similarly to the full inter-domain distance, showing good convergence at the end of the curve. This suggests an intuitive way of thinking of the local vs global points of view. The inter-domain distance (or similar magnitudes) are an average of many individual residue-residue distances, so that the fluctuation of the average–due to statistical reasons–must be much smaller than the individual fluctuations. Hence, the points of Yang and Min's studies are qualitatively different to the ones presented in this work, and should not be compared with them, as they represent different types of mechanisms in the protein.

**Long plateaus in DCs strongly suggest full convergence**. Is, then, a DC reaching a plateau a sufficient indication of convergence? In principle, the answer is no; several processes may have already reached equilibrium but some others, with a much longer time-scale, may still be developing. In this case, no plateau–no matter how long–would be enough to declare the trajectory converged. But modes of large flexible systems, like biomolecules, have a wide and overlapping distribution of characteristic times, as opposed to discrete and separated values. This is a consequence of the very large number of vibrational, rotational and torsional modes such a molecule posses (see Walton and Vanvliet's analysis[2] and references therein). Hence, reaching a clear (flat) plateau that extends for a significant time, strongly suggests that the modes related to the considered motion have already reached their decorrelation time. In Fig. 5, where analytical convergence was fully achieved (by design), this is clearly seen in the perfectly flat plateaus of the three individual sines and the exponential, and in the composite sin-HMLfreq function. The

other composite function, exp-sin-HMLfreq, does not reach such perfect convergence until a little later (not shown), but it can be seen that it is fastly approaching it. This analysis applied to Fig. 4 indicates that dialanine and GAAC are completely converged, while the others show a good general convergence, but still contain some not fully equilibrated processes. All of the latter systems suggest being close to the full convergence, with the exception, perhaps, of 3CL^pro–and elastase to a lower degree–which would require an extended simulation to make sure that last variation finally subsides.

**Convergence time is on the order of the period of the slowest oscillatory motions**. Panel A in Fig. 6 shows the inter-domain separation in PGK (excluding termini residues), where a period of oscillation of about 12 µs appears to be revealed, once the initial non-equilibrium fluctuations disappear, around 13 µs. Assuming a behavior like that showed in section II E, when analyzing the periodic components of the composite function (Fig. 5), we could estimate a convergence time, related to this oscillation, of $1.5 \cdot 12\,\mu s \sim 18\,\mu s$ which, converting to ps and taking its log, results in $\sim 7$, coinciding with the convergence time obtained from the actual calculation in Fig. 4.

A similar analysis helps to understand dialanine's DC: Table 4 shows the transition rates between the three main regions, B = (−100°, 32°), M = (32°, 112°), and T = (112°, 260°), defined in the $\psi$ domain. From them, it can be expected that the equilibrium of transitions will be reached once the corresponding three processes (transitions between regions) converge. And since the limiting rate is given by the MB/BM transitions–one per 9.9 ns–this implies an average period of 19.8 ns and a convergence time of 29.7 ns, resulting in a $\log(t.ps^{-1})$ of $\sim 4.5$. Just as in the previous case, for dialanine too, the convergence time found via the estimation of the largest period of motion results in the same order of magnitude as the rigorous calculation represented in Fig. 4.

**Is the given working definition of equilibrium good enough for an MD simulation of a protein? - theoretical considerations**. The dialanine study allowed us to understand that in the MD simulation of a small system, where one property such as the transition rate of $\psi$ is fully converged, there may be another one that is still very far from it. This is represented in Panel B of Fig. 6. Notice the curve for $\psi$ is very well converged, and the fluctuations in about 3/4 of the trajectory (after 7 µs) represent the minimal possible change (one event, 1/5 vs 1/6). On the other hand, the fluctuations in $\phi$ are extremely large, as from the local maximum at about 12.5 µs, the curve decreases by constant amounts, since there are no more observed transitions. Something similar happened with PGK, as previously shown, where the substructure of cluster 0 is not converged, as it was still changing at the end of the trajectory, but all other metrics (Fig. 2), including the DC, are clearly converged.

Both cases, dialanine and PGK, can be easily explained in terms of the previously discussed concept of the partition function and the irrelevance of requiring a full exploration of unlikely regions of the phase space, when what one wants is to explain or predict properties of the most probable regions. The same conclusion can be drawn from the concept of ergodicity and the ergodic hypothesis (of course, since the partition function and the ensembles that use it are a mathematical formalism that instantiates the ergodic hypothesis). Paraphrasing J. R. Dorfman in his lucid explanation of equilibrium[24], the ergodic hypothesis states that a mechanical system's trajectory in phase-space spends equal amounts of time, in equal volumes of that space. As a direct consequence, if a region of the phase-space is hardly reached in a

simulation, it must be due to its small volume, which makes it an unlikely region, and the values of some property there, thus, are also going to be unlikely. In other words, any system will spend most of its time in regions of the phase-space where the values of the interesting macroscopic properties are extremely close to their equilibrium values. A reasonable step from here is to think that practical uses of statistical mechanics only require a *soft* version of the ergodic hypothesis, that only applies to the low-dimensional phase-space that deploys the property under study. Apart from a few examples (transition rates in dialanine, sub-cluster structure in PGK, etc.) shown in this work, that support this soft version of the hypothesis, there exist analytical examples that display the same behavior of clear convergence of a given property (or in a given subspace of the phase-space), but not of other properties (the full space), such as the well-known Baker's transformation and the Arnold cat map[24].

In summary, following a *rigorous* physical definition of equilibrium and considering a *full equilibration* of the system could be inappropriate, since valuable information–such as $\psi$'s transition rate–would be lost, and in fact, the property may be perfectly well-defined in that low-dimensional phase-space. Hence, we posit our initial working definition of equilibrium in an MD is satisfactory.

## Conclusions

How should the working definition of equilibrium be applied? Given a specific property as a function of time (calculated from a simulation), one should calculate its DC and inspect it in order to decide if it ends in a plateau and, if it lasts for a *significant time*, conclude that all modes related to the subspace of the property have already converged. This conclusion is supported by the fact mentioned in section III (in *Long plateaus in DCs strongly suggest full convergence*), where it was discussed that large and flexible systems, like biomolecules, tend to present wide and overlapping distributions of characteristic times, so that when a plateau is found, it is likely that all modes have already reached convergence; and the longer the plateau, the more likely convergence was achieved.

Bear in mind that this is a reasonable, but not a rigorous, method. One could imagine a simple system, maybe a small molecule, where the distribution of decorrelation times could have some gaps; this seems more unlikely in large complex molecules, with many degrees of freedom. The decorrelation curve for GAAC (in Fig. 4) shows a mild counter-example: there is an apparent plateau between $\log(t \cdot ps^{-1}) \sim 5.7$ and 6.7. So, if the simulation lasted until, say, 6.5, one would be led to think that the property was converged. But having a longer trajectory, a small jump is observed at 7.0, indicating that the run was not converged at 6.5. It could be argued that this is, actually, not a counter-example, since between 5.7 and 6.7 the curve does not really represent a constant value, so it should not be taken as a plateau. In any case, this example is presented to show that clearly establishing convergence could be challenging.

To conclude, this work is suggesting that, from a biological point of view, where one is mostly interested in the function of a biomolecule and how it moves through its most likely conformations, the full convergence of a MD trajectory is not too relevant, as many specific properties of the system may be perfectly well-converged. For example, average properties, such as the energy, inter-domain distance or rotation, RMSD, RMSF, cluster distribution, do not depend much on the small contributions of low probability regions to the partition function, so they should be well described even if the full phase-space is not explored. On the contrary, there are properties that measure or depend directly on those contributions, like the transition rates to

**Table 3 Protein Srgence.**

| PROTEINS | Dialanine | Trp Cage | VHP | GAAC | Barnase | Elastase | PGK | 3CL$^{pro}$ |
|---|---|---|---|---|---|---|---|---|
| MD length [μs] | 20 | 22 | 22 | 44 | 15 | 12 | 30 | 100 |
| number of residues | 3 | 20 | 35 | 36 | 199 | 274 | 415 | 612 |

**Table 4 Dialanine's $\psi$ transition rates.**

| Transition type | TB/BT | TM/MT | BM/MB |
|---|---|---|---|
| Transition rate [ns$^{-1}$] | 1/0.9 | 1/7.5 | 1/9.9 |

T (top) stands for the $\psi > 112°$ region, B (bottom) for $\psi < 32°$, and M for the middle region. XY/YX is the rate for the combined X → Y and Y → X transitions. The bounds of the intervals were chosen so as to cover the three main populated regions in $\psi$, as observed in the free energy plot in Fig. 1, while keeping a significant amount of events in each of them (not less than 2%). Altering these bounds within reasonable limits does not change the order of magnitude of calculations based on them, shown later.

and from those regions, or any specific property of those areas (e.g., their energy). Along these lines, our findings show that all the properties that we studied in different systems, with varying number of aminoacids, tend to converge for trajectories on the order of tens of $\mu$s (some converge faster, even in the range of ns). In this sense, proteins appear to show equilibrium behavior for such time ranges.

Finally, it should be mentioned that the relevance of the studied convergence to in-vivo proteins is not direct and should be studied further, since they are not only in the presence of a static thermal bath, like in typical MD simulations, but are constantly driven by the inter-conversion of chemical, mechanical and thermal energy[25], that can drive the system out of equilibrium.

## Methods

**Metrics and systems.** We chose several metrics of convergence (Table 1), that test different aspects of the systems' behavior, including its global and local dynamical properties, as well as its structural properties. Notice that none of them explicitly tests thermodynamic aspects, like free energy, due to the complexity and long simulation times they require, but also due to the fact that they typically describe the equilibrium of the entire molecule, which is not what is needed in the present context. Nevertheless, thermodynamics appears implicitly in the probability distributions that we do test.

These metrics were applied to study the convergence of several proteins with a varying number of aminoacids, with the goal of probing the relation between the size of a protein and the time it needs to reach equilibrium. We also tested a few analytical systems, in order to better comprehend the nature of the convergent behavior (see Tables 2 and 3).

**Molecular dynamics.** Five of the systems mentioned in Table 3 (Dialanine, Trp Cage, VHP, Barnase and Elastase) were prepared and simulated by us, using the AMBER18[26] package of MD simulations. The initial structure of dialanine was built starting from its sequence, using the tleap program (part of AMBER), while the others were prepared, also with tleap, but starting from the structures corresponding to the following PDB codes: 1L2Y, 1YRF, 1BRS and 1PPF, respectively. After parametrizing the systems with the Amber force field ff14SB[27], solvating with TIP3P water and equilibrating the electric charge with counter-ions, the following protocol was applied: (1) 5000 cycles of steepest descent, followed by 5000 steps of conjugate gradient minimization; (2) 1 ns of heating, to 298 K, followed by 20 ns at constant temperature and pressure (1 atm) starting with soft harmonic

restraints $(k = 1 \, \text{kcal} \cdot (\text{mol} \cdot \text{Å})^{-1})$ on the protein, and slowly releasing them; (3) 10 ns restraint-free equilibration with constant temperature and volume (NVT ensemble); (4) several $\mu$s in the same ensemble (see Table 3 for specific lengths). All these simulations were run using the SHAKE algorithm, with a time-step of 2 ns, and with periodic boundary conditions with the Particle Mesh Ewald method, with a cut-off of the sums in direct space of 12 Å. All runs were performed using the GPU version of AMBER's PMEMD module.

The trajectories for the other three systems in Table 3 were taken from other sources: those of GAAC and 3CL$^{pro}$ are publicly available[13,28], while the trajectory of PGK was shared by its creators[3] upon request.

**Normal modes.** Normal mode analysis (NMA) is a well-known physical technique that, assuming a system is bound together by harmonic potentials, allows the extraction of the *natural* motions of a system, i.e., a set of independent motions that form an orthonormal basis of the vector space of physically accessible conformations of the system. In the last couple of decades, the method was successfully applied to proteins, under different versions, like Partial Component Analysis (PCA) extracted from MD simulations, and Elastic Network Models (ENM) using coarse-grained models of a protein, and using just the coordinates of one crystal structure (as opposed to using a whole trajectory from a simulation).

In this work, we model our systems using a particular version of ENMs, the Anisotropic Network Model (ANM)[29,30], where an N-residue protein is modeled in a coarse-grained way, each residue is taken as a single node (or pseudo-atom) located at the coordinates of the corresponding C$_\alpha$-atom, and the inter-residue interactions between them–only up to a cutoff distance of 12 Å–are represented by springs. The collective dynamics of the network is expressed by the *normal modes*, i.e., the 3N eigenvectors of the inverse Hessian matrix, $H^{-1}$, where the elements of $H$ are given by the second derivatives of the harmonic ANM potential.

ANM has been tested and used in many different proteins, and it has been shown to correctly describe their global motions which, in turn, have a strong effect in the function of the protein[14–18]. This has set the basis for an emerging paradigm: *structure encodes dynamics, which encodes function*. This suggests that, in biological terms, it makes sense to evaluate methods by looking at their resultant global motions. Consequently, we deem as reasonable to use the ANM results as a proxy of what should be obtained by other methods (MD) in order for them to be considered correct (or converged). In particular, a good way to estimate how much the modes obtained from MD overlap with those of ANM, is by using the Cumulative Overlap (CO) metric,

$$CO_i(j_{max}) = \sum_{j=0}^{j_{max} \leq 3N} P_j \cdot M_i, \qquad (5)$$

where $P_i$ and $M_i$ refer to normal mode $i$, obtained from PCA of an MD trajectory, and from ANM applied to a given conformation, respectively. This metric is used below to check for convergence of the MD trajectories; for each studied protein we calculate CO curves for different lengths of the trajectory (such that each one contains the previous one), expecting to see that, starting at some

particular length, they won't change anymore; this would indicate convergence of the CO, and suggest convergence of the trajectory.

**RMSF**. The well-known Root Mean Square Fluctuation (RMSF) of a atom $i$ is defined by

$$RMSF_i = \sqrt{\frac{1}{N}\sum_{j=1}^{N}\left(\mathbf{r}_i(t=j) - \langle\mathbf{r}_i\rangle\right)^2}, \tag{6}$$

where $\mathbf{r}_i$ represents its position vector, $N$ is the total number of steps in the trajectory, and $\langle\rangle$ is the time average. Just as in the $CO$ metric, it is expected that, as the MD length increases, the RMSF curves tend to converge to their equilibrium curve.

**Clusters**. Clustering is a method that classifies points in a distribution (e.g., protein micro-configurations) according to how they differ from each other. In this way, conformations in a trajectory can be divided into a few clusters, according to their similarity. In this work we used the *hierarchical agglomerative* and the *k-means* algorithms to perform the analyses.

**Auto-correlation functions**. The normalized ACF of a given property, R(t), such as a specific distance, is given by

$$C(\Delta, t) = \frac{C'(\Delta, t)}{C'(0, t)}$$
$$C'(\Delta, t) = \frac{1}{t - \Delta}\int_0^{t-\Delta}\delta R(t')\delta R(t' + \Delta)dt', \tag{7}$$

where $\delta R(t) = R(t) - \langle R\rangle$, $\Delta$ is the lag time, and $t$ is the total time of the trajectory.

The ACF determines how points in a time series relate, in average, to those occurring some lag time, $\Delta$, later. Hence, a function of $\Delta$ can be constructed, to show how that relation evolves with the lag-time. A large value of $C(\Delta, t)$, for a fixed $t$, indicates a self-similarity between the current and lagged points, while a small value indicates low correlations between them. In a dynamical process, like the time evolution of residues in a protein, it is expected that nearby measurements of a property will show a slow change; but that is not the case for (time) distant data points. Therefore, the $C(\Delta, t)$ should decrease as $\Delta$ increases. On the contrary, a random process would show no correlation between data points whether they are close or far away, making the ACF vanish for all $\Delta$. In this sense, the ACF can be thought of as a measure of the *memory* of the process.

## Data availability
The data that support the findings of this study are available from the corresponding author upon reasonable request. Movies of protein trajectories are available as Supplementary Movies 1–8, and pdb files containing first and last frames of each simulation are available as Supplementary Data files 1–16.

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

## Acknowledgements
I.J.G. acknowledges support from Agencia Nacional de Promoción Científica y Tecnológica, for grant PICTO-2021-CABBIO-00003.

## Author contributions
I.J.G. conceived and designed the project, and wrote the manuscript. F.O. and I.J.G. prepared and run the MD simulations, performed the analyses and interpretation of the data, and reviewed the manuscript.

## Competing interests

The authors declare no competing interests.
