## [Peer Review File · Communications Chemistry]

Reviewers' comments:

Reviewer #1 (Remarks to the Author):

This paper investigates the topic of convergence in regards to molecular dynamics simulations of biomolecules. The authors use a range of different methods to determine whether or not there is sufficient sampling to determine the value for an ensemble average. The authors final take is nuanced. They argue that the time it takes to reach equilibrium on a particular property depends on the property being measured. The motions of proteins, they demonstrate, occur across range of different timescales. Consequently, the simulator determining the ensemble average for a particular quantity should seek to investigate what length of simulation is required to determine the ensemble average of the property that interests them.

I think the paper should be published as it provides a number of useful ideas for investigating whether or not a simulation is converged. My one concern is that the amount of explanation of some of the figures is relatively short, given how much information is contained in the figure. For figure 2C, for instance, I don't really understand how the inset in the cluster graph was generated and why the results here are different from the results shown in the main figure. Was a clustering performed on only the final 17 us of the trajectory to generate this figure? In the words, is the result in the main part of this figure the results from clustering the whole trajectory and the inset the results from a clustering performed on the final 17 us only?

Reviewer #2 (Remarks to the Author):

This work analyzed of tens to a hundred microseconds long trajectories, of several systems with varying size, to probe the concept of equilibrium, and its physical and biological meaning. It could be published after following issues are considered.

1. The abstract need re-written. The authors need provide a bit more details of their main results, and explain the meaning if possible.
2. The introduction is hard to understand, the authors need explain their motivation and the background more clearly. Also, the conclusion is hard to understand. E.g. "s should not be described as being absolutely converged or not, but as being so in relation to specific properties."
3. Why did the authors discuss the velocity/energy distribution of their system to check if it has reached equilibrium?
4. The authors could proof read the manuscript carefully, since some parts/sentences are hard to read, or even misleading. e.g. "The trajectories for GAAC, PGK and 3CLpro were created by other authors; those of GAAC and 3CLpro are publicly available [8, 9], and that of PGK was shared by the authors [3] upon request."
5. the format of references need revision. E.g. why some journal papers show links while some do not? Ref 8 is a book, need show the location&published, etc.

Reviewer #3 (Remarks to the Author):

[Primary Question the paper aims to answer]

Have systems reached thermodynamic equilibrium at any given point within their trajectory? How can it be determined if the system has reached equilibrium?

[Approach to Question]

Unclear to me. They provide a thorough analysis using a host of well known systems but this paper does not provide a sound method for answering the primary question.

[Suggestion]

I think the paper would be better scoped if they re-defined the equilibrium definition they provide in Section 1. My suggestion is to define something like "epsilon-equilibrated" where they provide a probability bound on whether or not a system is equilibrated. If this was provided, this would be extremely useful because we may not ever know if a system is truly equilibrated unless we run it forever.

[Comment]

I think the paper should be addressing explicit thermodynamical properties, like free energy, instead of only implicit ones. This is an analysis paper and I feel like this is an appropriate request.

[Comment]

I wish there was a related works section because I am not completely familiar with all the background material and this would have been helpful for me.

[Comment]

The paper is generally well written but I feel that the major claims are weak. It is possible that many of the systems reported in literature are not equilibrated but the paper does not provide a sound method for at least giving a probability bound on equilibration. I believe that these results are of interest to the broader research community but many of us already have an informal knowledge of the contents of this paper -- this paper formalizes this knowledge. The paper would be very convincing to me if there was some kind of "epsilon-equilibrated" bound!

Convergence and Equilibrium in Molecular Dynamics Simulations

Dear Reviewers,

First we would like to thank you all for the very useful and thoughtful comments about the manuscript. We deeply considered all the opinions and suggestions, and tried to answer and correct accordingly. Below you can find the answers to each of the issues raised, where in black font are general comments, in red font specific questions and concerns, and in green font our answers.

But we would like to first comment on two of the issues mentioned by the 3rd reviewer. Although it is explained later, we just want to comment on two points related to the establishment of a method to assess some sort of a “probability of convergence”, and the use of free energy calculations:

1) Probability of convergence: We think that establishing such a method would be great, as one could test if his MD trajectory is enough or not, and assign a probability to the conclusions extracted from it. But one should not assume that such a method **MUST** exist. In fact, we think that it is not possible (or at least, we couldn't find a way) to calculate a non-arbitrary probability for this. But still, we propose a more qualitative method, based on some observations explained in the manuscript. Here, we just want to stress the idea that we cannot assume, by any means, that the quantitative method does exist.

2) Free energy: Free energy and entropy are the basic physical quantities studied in Statistical Mechanics, when talking about equilibrium. Nevertheless, we just didn't mention them in our previous manuscript. The reasons for that were two-fold: i) They are very computationally expensive properties to calculate, as they require many long simulations, apart from the original trajectory. And even if it were possible to do them, it is highly doubtful that the errors of the results are going to be small enough to clearly establishing convergence, as we need it here. ii) They are quantities that relate to the full protein, and not to individual properties, like the ones we wanted to study here (in terms of our language in this version of the manuscript, they are better for studying “full”, but not “partial” equilibrium).

With those two “preparation comments”, please find below the answers to the reviewers' comments.

Sincerely,

Ignacio J. General

Reviewer #1 (Remarks to the Author):

This paper investigates the topic of convergence in regards to molecular dynamics simulations of biomolecules. The authors use a range of different methods to determine whether or not there is sufficient sampling to determine the value for an ensemble average. The authors final take is nuanced. They argue that the time it takes to reach equilibrium on a particular property depends on the property being measured. The motions of proteins, they demonstrate, occur across range of different timescales. Consequently, the simulator determining the ensemble average for a

particular quantity should seek to investigate what length of simulation is required to determine the ensemble average of the property that interests them.

I think the paper should be published as it provides a number of useful ideas for investigating whether or not a simulation is converged. My one concern is that the amount of explanation of some of the figures is relatively short, given how much information is contained in the figure. For figure 2C, for instance, I don't really understand how the inset in the cluster graph was generated and why the results here are different from the results shown in the main figure. Was a clustering performed on only the final 17 us of the trajectory to generate this figure? In the words, is the result in the main part of this figure the results from clustering the whole trajectory and the inset the results from a clustering performed on the final 17 us only?

More information was added to the caption of some figures. The point raised by the reviewer in relation to Fig 2C was explicitly addressed in the caption, but also re-worded more clearly in the main text (and yes, the reviewer's interpretation about the inset was correct; it was a cluster analysis of the last 17 microseconds).

Reviewer #2 (Remarks to the Author):

This work analyzed of tens to a hundred microseconds long trajectories, of several systems with varying size, to probe the concept of equilibrium, and its physical and biological meaning. It could be published after following issues are considered.

1. The abstract need re-written. The authors need provide a bit more details of their main results, and explain the meaning if possible.

We re-wrote parts of the abstract, trying to show more details, particularly related to the meaning of our work.

2. The introduction is hard to understand, the authors need explain their motivation and the background more clearly.

We re-wrote the end of the 2nd paragraph to better express our puzzlement in relation to Hu's conclusion (some proteins do not reach equilibrium, even in long MD simulations on the order of tens of microseconds) and the lack of attention of the community to what we think is a huge issue for the MD community.

Also changed parts of the 4th paragraph, to clarify some points related to the "equilibrium" concept, and talked about why we study some structural and dynamical properties and not free energy or entropy. This is a technical description, with physics concepts, but we hope they are clear enough for the general reader.

And we also tweaked the definition of equilibrium a little, in order to make it more precise in terms of the concepts of equilibration of a specific property and equilibration of the full system (or partial and full equilibrium).

Also, the conclusion is hard to understand. E.g. "s should not be described as being absolutely converged or not, but as being so in relation to specific properties."

Rewrote that part, explaining more carefully that some properties extracted from a trajectory may be well converged (those whose values come from the most probable regions of the phase space), while others may not be (those that require exploration of low probability regions of the phase-space). Hence, full-convergence should not be a requirement when one only wants to extract properties like the energy, RMSD, clusters, global motions, etc, which fall in the first category (those properties that do not greatly depend on the low probability regions of the phase-space).

We also separated the previous “Discussion” section into a “Discussion” and a new “Conclusions” sections, and added some text in Conclusions to clarify the logic of the procedure we propose in order to check if some property is converged.

3. Why did the authors discuss the velocity/energy distribution of their system to check if it has reached equilibrium?

We are not sure what the reviewer is referring to. Maybe he is talking about our reference to the phase space? In that case, the answer is that in Physics, more precisely in Statistical Mechanics, it is shown that the convergence (or equilibrium) of a system is given as how thoroughly the so-called phase-space is covered by the simulation. This phase-space is a generalized space whose coordinates are the positions and velocities of each atom in the system. That is the reason why we mentioned velocity as an important variable.

Likewise, energy is also a magnitude that can be use to describe the equilibration of a system, although it is not as precise in this description, as the coverage of the phase-space. When a system is in equilibrium, its energy does not vary in time (there could be some relatively minor fluctuations, though).

If this is not what the reviewer was referring to, we will be glad to discuss further, upon his/her clarification.

4. The authors could proof read the manuscript carefully, since some parts/sentences are hard to read, or even misleading. e.g. “The trajectories for GAAC, PGK and 3CLpro were created by other authors; those of GAAC and 3CLpro are publicly available [8, 9], and that of PGK was shared by the authors [3] upon request.”

Not sure why the reviewer says that sentence is misleading but, anyway, we re-wrote the mentioned sentence to make it more clear: “The trajectories for the other three systems in Table \ref{table-systems2} were taken from other sources: those of GAAC and 3CL^{pro} are publicly available, while the trajectory of PGK was shared by its creators upon request. “ Also we re-read the whole paper and fixed other minor issues (in addition to the ones already mentioned).

5. the format of references need revision. E.g. why some journal papers show links while some do not? Ref 8 is a book, need show the location&published, etc.

Fixed.

Reviewer #3 (Remarks to the Author):

[Primary Question the paper aims to answer]

Have systems reached thermodynamic equilibrium at any given point within their trajectory?
How can it be determined if the system has reached equilibrium?

[Approach to Question]

Unclear to me. They provide a thorough analysis using a host of well known systems but this paper does not provide a sound method for answering the primary question.

Our intention was not to describe a method to check for convergence, but to asses convergence in some specific multi-microsecond long MD simulations, in order to contrast the claims in Hu’s paper. But we understand we were not too clear about the final goal, so we reworded parts of the abstract, introduction and conclusions, to make this more clear. And we also added a procedure (in the Conclusions section) that can be followed to check convergence of specific properties.

[Suggestion]

1. I think the paper would be better scoped if they re-defined the equilibrium definition they provide in Section 1. My suggestion is to define something like "epsilon-equilibrated" where

they provide a probability bound on whether or not a system is equilibrated. If this was provided, this would be extremely useful because we may not ever know if a system is truly equilibrated unless we run it forever.

We did actually consider establishing some kind of “equilibrium probability” at some point in

our study, but desisted to pursue it because we now think there is no reasonable (non-arbitrary) way of defining it. Here is a simple example that shows the futility of it. Consider a system whose real but unknown potential energy landscape (along some given reaction coordinate x) is given by a function like the “Mexican-hat” (see figure). The green curve is the potential energy, and the red line is the total energy of the system. The probability to find the system in each x position (horizontal axis) is proportional to the difference between the heights of the green and red curves at that x . Thus, it is very likely to find the system around $x=-1$ or $x=1$, while it is highly unlikely to find it at $x=0$, and the system will not easily transit from the negative to the positive x side, or vice-versa. Still, the difference at 0 is very small but not zero, and thus the system will eventually move to the other side. So, if a simulation is started with the system located at a negative x , it will probably stay there for a very long time, maybe for the whole simulation. Since it stayed in the negative side, appearing very stable, we will assign a high probability to the fact that the simulation is converged, and will conclude that x is always negative. But this is simply wrong. The probability of having achieved convergence should have been low, since the real potential is symmetric and, in a very long simulation, the system will be equally found on both sides of x . What we see from this example is that establishing a probability estimate may only lead to a false sense of certainty and thus, we consider it best to accept that without already knowing the energy profile of the system, it is—in general—not possible to calculate or even estimate a probability of convergence.

The only hint at a reasonable probability assignment that we could think of, and that we mentioned in the manuscript (“Long plateaus in DCs strongly suggest full convergence” subsection), was that related to the plateaus in decorrelation curves. As explained in the text, large and flexible molecules (like proteins) tend to have a wide and overlapping distribution of characteristic times (due to their many degrees of freedom), and it is not likely that there will be a large gap in their distribution. Finding a plateau in a decorrelation curve, would then mean that no mode reached equilibrium during the time of the plateau. And if the plateau extends for a significant time and suddenly ends, it would mean that there was a gap in the mode distribution, contradicting what we said before about biomolecules. Hence, looking for long plateaus in decorrelation curves seems like a sensible guide to assert convergence. But again, this is just a qualitative guide, since we don’t find any valid reason to choose specific time-lengths to infer a probability. Should the plateau extend for 100 ns in order for the run to be converged? Or 1 μ s? Or 10 μ s? Going back to the Mexican-hat example, the extension of the plateau is a function of the difference between green and red curves at $x=0$. If the difference tends to 0, then the time would tend to infinity. And notice that in the manuscript we even have a counter-example to this method; the decorrelation curve for GAAC shows an apparent plateau between $\log(t) \sim 5.7$ and 6.7 , but then there is small jump at 7. This is not really a perfect counter-example, since the first plateau was not too horizontal, but still, it serves to show the difficulty in establishing a good quantitative method.

We included some of the previous text in the manuscript. The Discussion section of the previous manuscript version was broken into two parts, right where the paragraph starting with “In summary, following a rigorous physical definition of equilibrium...” starts. And there we

added a sort of a guide one can use to check for convergence. Again, it is not a precise prescript, but more of a qualitative guide along the lines discussed above.

[Comment]

2. I think the paper should be addressing explicit thermodynamical properties, like free energy, instead of only implicit ones. This is an analysis paper and I feel like this is an appropriate request.

Free energy and entropy are, perhaps, the properties most related to equilibrium. Hence, it would be ideal to run calculations and look at their behavior. Unfortunately, these calculations are the most expensive ones in MD. Due to the level of precision we want to have in this work, the use of fast and relative low-quality free energy methods, like MM-PBSA, MM-GBSA, LIE, etc, are out of the question, since we believe they are not precise enough to measure equilibration (due to approximations of the methods, the resulting errors, in the order of several kcal/mol, are too large). On the other hand, much more precise methods, like Thermodynamic Integration, Hypothetical Scanning MD, etc, probably have the precision needed, but they are extremely expensive. With the vast experience of one of the authors in free energy calculations, we estimate that in order to build a convincing picture with this type of methods, we will need to do, at the very least, ten individual calculations, each to get the free energy at a given time. Each full calculation would represent, at least, a month-long calculation. And this should be replicated for all considered systems. A quick and very conservative estimation results in computations that will take, at least, 6 months, dedicating all our resources to this. Unfortunately, this is far from our current possibilities. And even if we could do it, it is doubtful that we could clearly decide on convergence, due to the size of the errors in typical free energy calculations (in my experience, not less than 1 kcal/mol).

With that being said, we are planning a follow-up project, where we intend to explore free energy calculations for only one or maybe two of the systems. But due to the long times needed and the issue of the error, as mentioned, that will be a project by itself.

But, although free energy is somehow the ideal metric when talking about equilibrium, is it really ideal in our case? We don't think so. Free energy can provide information about the stability/convergence of the whole protein, including all internal motions. Only when all of them are equilibrated the free energy will show convergence. But as we show in the manuscript, biologically speaking we are not interested in the full convergence, but in the convergence of specific, biologically relevant, properties. An inter-domain oscillation may be functionally very important, while an inter-atomic one may not. Hence, a free energy calculation will not show convergence, while functionally, the system might be in "partial equilibrium". In the 4th paragraph of the "Introduction", we added an explanation of why free energy is not a useful metric when looking at this partial equilibrium. We also mentioned it in the 1st paragraph of "Methods".

Note: There are some ways to calculate free energy of parts of the system (e.g., one particular domain)—although they are somewhat arbitrary—but that will just tremendously increase the computational cost of the calculations, since one would need to do this for several different parts of the molecule. This we will explore in that follow-up project.

[Comment]

3. I wish there was a related works section because I am not completely familiar with all the background material and this would have been helpful for me.

We are not sure what specific background material the reviewer is referring to (Methods? Previous results? Physical concepts, like the partition function?). But would be glad to add more

information on any of those or related topics, if he/she explains what material may help the reader.

Related, we did add some more info about the physics of equilibrium, in the 4th paragraph of the Introduction.

[Comment]

The paper is generally well written but I feel that the major claims are weak. It is possible that many of the systems reported in literature are not equilibrated but the paper does not provide a sound method for at least giving a probability bound on equilibration. I believe that these results are of interest to the broader research community but many of us already have an informal knowledge of the contents of this paper -- this paper formalizes this knowledge. The paper would be very convincing to me if there was some kind of "epsilon-equilibrated" bound!

As previously mentioned, a probability bound for convergence would definitely be great. But unfortunately, we just don't see a good way of doing it, with the exception of the "long plateau" idea. But this is a qualitative test.

But reading the reviewer's comment, I think the previous version of the manuscript was probably not clear enough about the fact that our goal was mostly to contrast the conclusion in Hu et al, that some proteins are non-equilibrium for time-spans of seconds. This conclusion implies that MD simulations are mostly meaningless. The conclusion of our work is that this is not the case, since the functional dynamics of proteins has faster convergence times: normal modes have been shown to be fundamental in explaining function of proteins, and normal modes are converged (as showed in our work), along with other important properties usually studied in MD, like clusters, inter-domain separations, etc.

With all the changes mentioned in this letter, we think the central goal of the paper should now be much more clear.

REVIEWERS' COMMENTS:

Reviewer #1 (Remarks to the Author):

The authors have appropriately addressed my concerns.

Reviewer #2 (Remarks to the Author):

it could be published now

Reviewer #3 (Remarks to the Author):

Thank you for your revised manuscript and your rebuttal letter. I feel that the revised manuscript represents a significant improvement over the original submission and I am much more confident in this submission. I feel that my initial concerns were address and this manuscript is suitable for publication.